# The Impact of a Community Pharmacy-Led Deworming-Related Counselling Service: An Interventional Study in a Low-to-Middle Income Country

**DOI:** 10.3390/tropicalmed10080215

**Published:** 2025-07-30

**Authors:** Amira B. Kassem, Ahmad Z. Al Meslamani, Mohamed AbdElrahman, Nadia Al Mazrouei, Sherouk M. Okda, Noha A. El-Bassiouny, Asmaa Abdel-hamed Hamedo, Doaa Abdelrazek Shaban, Dina Fathy Elsmadessy, Ammena Y. Binsaleh, Asmaa Saleh, Hebatallah Ahmed Mohamed Moustafa

**Affiliations:** 1Department of Clinical Pharmacy and Pharmacy Practice, Faculty of Pharmacy, Damanhour University, Damanhour 22514, Egypt; sherouk.okda@pharm.dmu.edu.eg (S.M.O.); noha.el.bassiouny@pharm.dmu.edu.eg (N.A.E.-B.); 2College of Pharmacy, Al Ain University, Abu Dhabi P.O. Box 112612, United Arab Emirates; ahmad.almeslamani@aau.ac.ae; 3AAU Health and Biomedical Research Center, Al Ain University, Abu Dhabi P.O. Box 112612, United Arab Emirates; 4College of Pharmacy, Al-Mustaqbal University, Babylon 51001, Iraq; mohamedmahmoud@uomus.edu.iq; 5Clinical Pharmacy Department, Badr University Hospital, Faculty of Medicine, Helwan University, Helwan 11795, Egypt; 6Department of Pharmacy Practice and Pharmacotherapeutics at College of Pharmacy, University of Sharjah, Sharjah P.O. Box 27272, United Arab Emirates; nalmazrouei@sharjah.ac.ae; 7Beheira Specialized Children’s Hospital, Beheira 11588, Egypt; hamedoasmaa77@gmail.com; 8Sanhour Health Unit, Damnhour, Beheira 22516, Egypt; doaaabdelrazek58@gmail.com; 9Vaccination Department, Elmahmoudeia Health Unit, Elmahmoudeia, Beheira 22753, Egypt; dinafathy7676@gmail.com; 10Department of Pharmacy Practice, College of Pharmacy, Princess Nourah bint Abdulrahman University, P.O. Box 84428, Riyadh 11671, Saudi Arabia; aysaleh@pnu.edu.sa; 11Department of Pharmaceutical Sciences, College of Pharmacy, Princess Nourah bint Abdulrahman University, P.O. Box 84428, Riyadh 11671, Saudi Arabia; asali@pnu.edu.sa; 12Clinical Pharmacy and Pharmacy Practice Department, Faculty of Pharmacy, Badr University in Cairo, Cairo 11829, Egypt

**Keywords:** patient counseling, parasitic infection, deworming, helminthiasis, community pharmacists, hygiene

## Abstract

**Background:** Since the current increase in antimicrobial resistance globally, parasitic infectious diseases have become a greater public health crisis than ever before and an absolute priority. The Egyptian community pharmacist, as a health care provider and advisor, has a potential role to play in deworming. **Objective**: To evaluate the outcomes of community pharmacist-led deworming-related counseling services. **Methods:** A prospective randomized controlled study was conducted in Damanhur, Behera, Egypt. The intervention group received community pharmacy counseling, and the control group received the usual care. Both groups were monitored for 12 months for recurrence evaluation. **Results:** A total of 173 patients were included, of whom 96 (55.5%) received patient counseling. The types of infection included *Oxyuris* (44.5%), *Entamoeba histolytica* (28.9%), Ascaris (8.7%), *Entamoeba Cyst* (8.7%), *Giardiasis* (4.6%), *Schistosomiasis* (2.9%), and *pinworm* (1.7%). A total of 119 participants (68.8%) reported a need for dose repetition. Personal hygiene practices were reported by 71 participants (41%). Recurrence of infection was observed in 101 cases (58.4%). Patient counseling was significantly associated with lower recurrence rates and higher rates of personal hygiene application (*p* < 0.001). Multivariate logistic regression analysis revealed that patient counseling and personal hygiene measures were the only significant predictors of infection recurrence rate. **Conclusions:** Community pharmacists’ deworming-related counseling had positive behavioral and clinical outcomes.

## 1. Introduction

### 1.1. Introduction to Parasitic Diseases

Parasitic diseases, such as protozoal infections and worms, are among the neglected tropical diseases designated by the World Health Organization (WHO) because their surveillance, prevention, and treatment are often neglected [1,2,3]. Parasitic diseases continue to increase global mortality and the economic burden [4,5,6,7], especially in low-to-middle-income countries (LMICs) among illiterate populations and settings with poor sanitation [4,8,9,10].

Although there is a global dependence on drug donations of chemotherapy for neglected tropical disease (NTD) control [11,12], endemic countries still have insufficient breaks of transmission [13,14]. The development of novel antiparasitic drugs and the use of drug combinations need to be envisioned to address the potential resistance [15,16,17] that may be aggravated by using generic substandard anthelmintic drugs [18].

Parasitic disease management over the past few decades posed a tremendous challenge in Egypt [8,19]. In the era of “One Health” and emerging resistance to all antimicrobial agents, Egypt intends to combat parasitic infections and develop effective prevention strategies, where the Egyptian community pharmacist has a potential role to play [19,20,21,22,23]. The effectiveness of available deworming drugs is compromised by emerging drug resistance, cross-resistance, and lower-than-needed novel drug development [15,18]. The WHO Road Map for NTDs supports endemic countries to implement mass anthelmintic treatment, especially for children and child-bearing-age women (who are not in the first trimester of pregnancy) [14,24,25,26], and to overcome barriers to diagnosis and prevention of parasitic infections [27].

### 1.2. Parasites

Parasites are microorganisms that live on or inside a host and benefit at the expense of this host [28]. Medically relevant parasites include a diverse array of microorganisms, such as protozoa, helminths, and ectoparasites [29]. They cause debilitating disability and are often stigmatized [30]. They are particularly common in poor communities [8,19]. Healthcare providers should consider the risk of infection in subjects living in or traveling to endemic areas [31].

### 1.3. Protozoa

Protozoa are microscopic unicellular organisms that contribute enormously to the infectious disease burden worldwide [2,32,33]. They include *Giardia intestinalis*, *Cryptosporidium* spp., and *Entamoeba histolytica* [21]. Protozoal infections are highly prevalent in LMICs, especially in rural areas and among those living in crowded urban environments. Causes of protozoal infections include poor sanitization, lack of clean water access, low socioeconomic conditions, and living close to carriers [2]. Protozoan infections range from asymptomatic to fatal. The health of the host plays a role in the spread of infection. Carriers’ defense systems can control, but not eliminate, an infection [15]. Dormant cysts in the life cycle of protozoa survive in difficult conditions for a long period with no oxygen, water, or nutrients [15]. The effectiveness of antiprotozoal drugs is diminished due to resistance development, the emergence of cross-resistance, and the lack of vaccines [15,21].

*Entamoeba histolytica* is an invasive pathogenic intestinal amoeba protozoan [34]. Amoebas are protozoa that use their pseudopodia for movement and may cause dysenteric liver abscesses [32,33]. Many of the individuals infected with *Entamoeba histolytica* are asymptomatic [35,36]. Factors related to the host, such as gene expression, play a role in invasive amebiasis [37,38]. Invasion may be accompanied by inflammatory tissue damage, and the leptin signalling activity may confer resistance to *Entamoeba histolytica* [39]. In addition, oxidative stress may enhance the parasite viability by suppressing gene expression [40]. Genes associated with *Entamoeba histolytica*’s virulent phenotype include STIRP, which plays a role in the interaction of the amoeba with

Target cells [41]. Amoebiasis can be diagnosed immunologically using a patient’s stool or blood [42] or by molecular biology techniques, which can differentiate the pathogenic entamoeba from other harmless species that look similar under microscopy [34,43]. Five-NI drugs, such as metronidazole, tinidazole, secnidazole, and ornidazole, are the mainstay of amoebiasis treatment [21]. Intestinal amoebiasis treatment failure occurs mostly due to recurring infection, rather than drug resistance [44,45]. However, lower drug susceptibility was also reported [46].

### 1.4. Heliminths

Helminths are multi-cellular parasitic worms [25,29]. They include nematodes (roundworms, e.g., *Ascaris lumbricoides*, and *oxyuris*), trematodes (flatworms/flukes, e.g., *Schistosoma*), and cestodes (tapeworms) [5,8,25,47]. Soil-transmitted helminthiasis (STHs), or geo-helminths, include hookworm (*Necator americanus* and *Ancyclostoma duodenale*), roundworm (*Ascaris*), and whipworm [25,48,49]. Contaminated soil plays a big role in their transmission. Eggs are deposited through feces in the local environment via open defecation, using soil for cleansing hands, or poor hygiene after defecating. The eggs then undergo molting in the soil before re-entering the human host. Helminths cannot reproduce within their hosts [25,50]. SHT transmission occurs via skin penetration, accidental ingestion, or a vector bite [25].

Most of the worms live in warm and moist geographic regions (tropics and subtropics) [51]. Countries with high standards of environmental and home hygiene, such as Japan, are free from worm infestations nowadays [52]. Most worm infections are mild and asymptomatic. Moderate to heavy helminth infestations may be associated with nutritional compromise (mainly fats and vitamins), enteropathy (diarrhea, abdominal pain, malaise, weakness, anorexia), growth faltering, impaired cognitive function, and poor academic performance among children [4,25,53,54,55].

Molecular-based approaches, such as PCR (both traditional and real-time), are newer techniques used to detect several helminth infections [56]. Commonly used deworming drugs include albendazole, mebendazole, and praziquantel. They are safe and efficacious against a variety of worm infestations [48]. Stool microscopy is used to identify helminths [27]. Improving the health outcomes for individuals suffering from these illnesses is multifaceted. Improved access to water, sanitation, and hygiene is also recommended in the sustainable development goals [57].

### 1.5. Preventive Measures Against Parasitic Infections

The adoption of effective preventive measures and the provision of individualized therapy are highly warranted [27]. The key factors underpinning parasitic infections are poverty, overcrowding, poor water supply, inadequate food hygiene, open defecation, and warm climate. Key drivers of improving personal and environmental hygiene, such as eliminating open defecation, proper hand washing after toileting, proper sanitation, changing underwear regularly, cutting nails, proper washing and drying of clothes, bed sheets, and toys, maintaining a clean kitchen, and using food utensils for eating, should be promoted while counseling the members of the family. Family members and housekeeping staff should receive prophylactic treatment to prevent person-to-person transmission [27,49,51].

### 1.6. Challenges of Parasitic Infections Management and Role of the Pharmacist

Parasitic infections often lead to physical, social, and economic burdens [58]. Adherence to deworming drugs is usually low due to their severe adverse reactions [59]. The resistance to deworming medications is widespread and increasing [15,21,60]. Because many of the drugs used are uncommon [12], coordination between the physician and the pharmacist is crucial.

The pharmacist is an integral part of the healthcare workforce [61,62,63,64]. Community pharmacists have the advantage of the flexibility of appointments, as patients receive one consultation rather than have to visit multiple providers to receive the same service [65,66,67]. The expansion of community pharmacy-based services has driven the majority of the public to perceive them as public health actors [31,58,61,68,69]. Because of their experience and knowledge, community pharmacists can treat minor ailments and ensure optimal care [67,70,71,72,73,74]. The community pharmacist has a pivotal role in monitoring, managing antimicrobial resistance, informing patients, and avoiding deworming drug misuse [60,75].

Parasitic diseases are a challenging specialty, requiring up-to-date knowledge of drug resistance and both international and local health regulations. Because tropical medicine is mainly focused on preventive health care, the patient’s health will depend on the healthcare practitioner’s expertise and proficiency in providing counseling. The community pharmacist can also reinforce the use of recommended hygiene measures [76,77,78,79]. The expansion of the scope of practice of community pharmacists with training would further improve patients’ access to optimal care at community pharmacies. Of note, in some countries, pharmacists with certain qualifications can independently prescribe within the scope of their specialty [67,80,81]. However, without an adequate competent workforce running an already busy pharmacy, it would be challenging to maintain the workflow and incorporate the new specialized service into their workload [67,82].

Notably, intensive collaboration between the prescribers and community pharmacists is needed to ensure optimal care, and providing the community pharmacists with the required appropriate training can ensure the proper implementation of WHO health guidelines and accelerate progress toward sustainable development’s healthcare goals [83]. Including the pharmacist in an interprofessional team helps to drive better outcomes in endemic regions [75,84]. Based on the above, the community pharmacist has a potential role to play in advising and counseling patients with parasitic infections. To the best of our knowledge, there has not been a study that evaluates the community pharmacists-led services regarding the management of parasitic diseases. Therefore, this study aims to evaluate the impact of community pharmacist-led patient counseling services regarding applying proper personal hygiene, preventing the recurrence of parasitic infection, and reducing the need for dose repeat

## 2. Patients and Methods

### 2.1. Study Design and Setting

This was an open-label randomized clinical trial (RCT) involving 173 patients from Damanhur, Behera, Egypt, collected from community pharmacies. Participants were randomly assigned to either the intervention group (*n* = 96) or the control group (*n* = 77), resulting in an allocation ratio of approximately 1.25:1. There were no exclusion criteria for patients. The data for every patient was collected in a Google form, including sex, age, weight, type of infection, and drug administered. The intervention group received one face-to-face counseling session outside the dispensing area in a private consultation area in the pharmacy during a specific appointment to ensure the quality of care and two mobile-phone follow-up counseling sessions about drug use, along with non-pharmacological hygiene measures. The primary endpoint of this study is the reduction in the recurrence rate of parasitic infections after 12 months. At the end of this 12-month study, both the need for a dose repeat and adherence to personal hygiene were directly assessed by questioning the patient.

### 2.2. Recruitment

Community-pharmacist-led services were promoted by contacting the patients, using posters and flyers, and referral by local physicians. A standardized training program was conducted for the pharmacists involved in the intervention group prior to the study. This training focused on consistent delivery of counseling content, including drug use, hygiene practices, and adherence strategies, based on a prepared counseling protocol. To avoid bias, different pharmacists were assigned to the intervention and control groups. Those providing counseling services were not involved in dispensing medications to the control group participants.

### 2.3. Randomization

An online tool for randomization was used to reduce selection bias [85].

### 2.4. Intervention

In addition to the conventional community pharmacist services (supply of medications), the community pharmacist gave the proper patient counseling about the prescribed anthelminthic, including the correct dose and duration. They also provided patients with information on the importance of non-pharmacological measures, such as personal hygiene. To reinforce messages and monitor progress, two telephone calls were scheduled: in Call 1 (week 2), the community pharmacist verified treatment initiation, checked for adverse effects, solved practical barriers (e.g., reminder tools), and reiterated key hygiene behaviours. In Call 2 (week 8), the community pharmacist assessed completion of therapy, reviewed persistence of hygiene practices, screened for recurrent symptoms, and reminded patients of the 12-week stool-analysis appointment. Of the 96 participants in the intervention group, 90 (93.8%) completed both calls, 4 (4.2%) completed one call, and 2 (2.1%) could not be reached despite three attempts and, therefore, received no follow-up call.

These services were provided in a designated patient counseling area inside the community pharmacy. After that, the patients were followed up to promote adherence to medication and hygiene measures and to determine the impact of patient counseling on reinfection. Hygiene practices were measured at two time points: at baseline (pre-intervention) and at follow-up (12 weeks post-intervention). Personal hygiene practices were assessed using a structured, self-administered checklist comprising eight items, including:washing hands with soap and water after using the toilet;washing hands with soap and water before and after eating;trimming fingernails regularly;drinking clean, safe water (filtered/boiled/bottled);changing and washing bed sheets regularly;washing fruits and vegetables thoroughly before eating;defecating in a sanitary toilet, not performing open defecation;educating my children/family members about handwashing and hygiene

Participants were asked to indicate whether they practiced each behavior consistently (“Yes”/”No”) during the study period. The score was then converted into a binary outcome (adequate vs. inadequate hygiene) for analysis. The pharmacists provided counseling services at a dedicated time outside the conventional dispensing time under a pharmacist–physician collaborative protocol. Recurrence was measured based on self-reported symptoms and repeated stool analysis.

Outcome measures are defined as follows: recurrence, defined as self-reported gastrointestinal symptoms consistent with the baseline infection and laboratory confirmation of the same parasite species within 12 months; adequate hygiene, defined as affirmative (“Yes”) responses to ≥6 of the 8 checklist items at the 12-week assessment; and dose repeat, defined as receipt of an additional anthelmintic dose after completion of the initial course, documented in pharmacy records or self-report. These binary outcomes (Yes/No) were used in all subsequent analyses.

### 2.5. Sample Size Calculation

The sample size for this trial was calculated to determine the effect of patient counseling on adopting personal hygiene measures and reducing the recurrence rates of parasitic infections in Egypt. The primary endpoint of this study is the reduction in the recurrence rate of parasitic infections after 12 months.

To estimate the necessary sample size, we adopted Rosner’s equations, in which we assumed a two-sided significance level (α) of 0.05 and a power (β) of 80% (power = 0.80). Based on previous studies, the recurrence rate of parasitic infections without intervention is approximately 30% (p1 = 0.30). We aim to achieve a clinically meaningful reduction of 10% in recurrence rates with counseling, bringing the rate down to 20% (p2 = 0.20). The calculated sample size required for each group (intervention and control) to achieve a 10% reduction in the recurrence rate of parasitic infections with 80% power and a 5% significance level is approximately 62 participants. Therefore, a total of 124 participants (62 in each group) would be required for this study. However, in this study, we were able to secure 77 controls and 96 participants in the intervention group.

Rosner’s equations:$$N1={\left\{\right.\;Z1-\frac{\alpha }{2}*\sqrt{\bar{\bar{p}*\bar{q}*(1+\frac{1}{k})}}\left.\;+z1-\beta^{*}\;\sqrt{\bar{p1*q1+(\frac{p2*q2}{k})}}\right\}}^{2}/\Delta^{2}$$
q1=1−p1
q2=1−p1
$$\;\bar{p}=\frac{p1+kp2}{1+k}$$
$$\;\bar{q}=1-\bar{p}$$
$$N1=\left\{\right.1.96*\sqrt{\bar{0.2*0.8*(1+\frac{1}{1})}}+0.84^{*}\left.\sqrt{\bar{0.3*0.7+(\frac{0.1*0.9}{1})}}\right\}\;2/0.2^{2}$$
N1=62
$$\;N2=k^{*}N1=62$$
where

p1 and p2 represent the proportions (incidences) of groups 1 and 2, respectively.

Δ denotes the absolute difference between the two proportions, calculated as |p2 − p1|.

n1 is the sample size for group 1.

n2 is the sample size for group 2.

α indicates the probability of making a Type I error, typically set at 0.05.

β indicates the probability of making a Type II error, commonly set at 0.2.

z represents the critical Z value corresponding to a given α or β.

K is the ratio of the sample size of group 2 to that of group 1.

### 2.6. Statistical Analysis

SPSS version 27 was used for data analysis. All variables were categorical and presented as numbers with percentages. In addition to descriptive analysis, the chi-square test was used to test differences in recurrence rates across other variables. Logistic regression was conducted to evaluate the associated factors with the infection recurrence rate. A *p*-value of less than 0.05 was considered significant.

## 3. Results

The study included 173 patients, of whom 51 (29.5%) were over 18 years old (Table 1). Regarding gender, females represented a higher proportion with 92 participants (53.2%). The types of infection observed were predominantly *Oxyuris* with 77 cases (44.5%), followed by Entamoeba with 50 cases (28.9%), *Ascaris* with 15 cases (8.7%), *Entamoeba Cyst* with 15 cases (8.7%), Giardia with 8 cases (4.6%), *Schistosomiasis* with 5 cases (2.9%) and *Pinworm* with 3 cases (1.7%). Personal hygiene practices were reported by 71 participants (41%). Among the participants, 96 (55.5%) received patient counseling. Regarding dose repetition, 119 participants (68.8%) reported a need for dose repetition, while 54 participants (31.2%) did not. Finally, recurrence of infection was observed in 101 cases (58.4%). As shown in Figure 1, Albendazole (71, 41.04%) was the most commonly used drug, followed by Metronidazole (59, 34.10%) and Flubendazole (23, 13.29%).

The findings showed that gender and dose repetition were not statistically significant factors affecting recurrence rates (*p* < 0.05). The recurrence of parasitic infections was highest in children aged 6–11 years (*p* < 0.05). Patients infected with *Entamoeba* and *Oxyuris* showed relatively high recurrence rates, whereas those with *Schistosomiasis, Giardia*, and *Pinworm* infections had lower or no recurrence (*p* < 0.05). Personal hygiene practices were significantly increased (*p* < 0.001), and recurrence rates were significantly reduced (*p* < 0.001) with community-pharmacist-provided patient counseling (Table 2 and Table 3).

Those not adhering to personal hygiene had a much higher recurrence rate (83.3%) compared to those practicing hygiene (22.5%). Similarly, patient counseling showed a significant impact, with 93.5% recurrence in patients who did not receive counseling versus 30.2% in those who did, as shown in Figure 2.

Logistic regression was conducted to assess factors affecting the recurrence rate. In univariate logistic regression analysis, the age group > 18 years showed statistically significant lower odds of recurrence (OR = 0.203, *p* < 0.001), as did the 0–5 year group (OR = 0.389, *p* = 0.037), compared to the reference age group of 6–11 years. Moreover, patient counseling was significantly associated with a lower risk of recurrence (OR = 0.03, *p* < 0.001). Similarly, adherence to personal hygiene measures also reduced the risk of recurrence (OR = 0.058, *p* < 0.001).

Regarding the type of infection, none of the specific parasites (including *Entamoeba*, *Ascaris*, *Oxyuris*, *Giardia*, and *Pinworm*) showed a statistically significant association with recurrence in the univariate and multivariate models when compared to the reference group (Schistosomiasis), despite some showing high odds ratios. Other variables, such as gender and repeated doses of medication, were not significantly associated with infection recurrence in either analysis.

After adjusting for age, gender, and infection type in the multivariate logistic regression model, patient counseling and personal hygiene measures were the only significant predictors of the infection recurrence rate (*p* < 0.01), as shown in Table 4.

## 4. Discussion

In this prospective open-label randomized control study of adults and children from a low-to-middle-income country (LMIC) receiving counseling for deworming, we found that community pharmacist-provided patient counseling was significantly associated with lower recurrence rates and higher rates of application of personal hygiene practices.

Although there was a safe water source in the region where the study was performed, other factors may promote parasitic infections, such as poor hygiene measures, consuming raw food, and walking barefoot [86,87]. *Oxyuris* and *Entamoeba* were the most prevalent infections in our study sample. The parasite species identified in two previous studies included *Ascaris*, *Oxyuris*, *Entamoeba*, and *Giardia* species [88,89].

Indiscriminate use of antiparasitic drugs is a matter of concern in LMIC, regardless of the socioeconomic or educational level of the population [90,91]. More than half of our sample were females. We did not find a significant gender-based difference concerning the rate of recurrence. Girls and women are more difficult populations to reach in LMICs [92].

Nearly two-thirds of our patients were pediatric and most of the infections were helminth infections. Helminth infections are more frequent in pediatric patients and may lead to resistance to medication and reinfection in adulthood [34]. Nearly 3% of patients in this study were infected with *Schistosoma*, which is highly prevalent in the Beheira governorate [8]. The recreational swimming behavior of children in canals in rural areas has greatly decreased over the last decades, which has decreased the spread of infection [93].

Drug administration can reduce morbidity. However, it is unlikely to interrupt transmission by itself due to reinfection [94]. Reinfection with helminth after treatment is the norm [95,96]. The use of any antimicrobial agent, including deworming medications, should be meticulously monitored to minimize the emergence of resistance [21,59]. This highlights the role of supporting drug administration with patient counseling and improved hygiene in ensuring the successful elimination of parasitic infections [97]. Supporting a shift from control to elimination is a challenge that requires strategic research and action [98]. 

Community pharmacists are in a good position to provide healthcare services because of their accessibility [31]. In addition to their roles as drug professionals, community pharmacists can play a role in the management of infectious diseases through their healthcare advice and patient counseling activities [83]. Previous studies reported high levels of patient satisfaction with pharmacist-led healthcare services [99,100,101]. It is the responsibility of the community pharmacist as a health promoter to inform, educate, and counsel patients about the management of these parasitic diseases and the risks associated with drug misuse and drug resistance [102,103]. We found that community pharmacist-provided patient counseling significantly reduced parasitic infection recurrence rates, in line with a previous study [104]. Increasing patients’ awareness of community pharmacy services and promoting collaboration with physicians was previously recommended in the literature [105]. The community pharmacists in this study provided counseling services under a pharmacist–physician collaborative protocol.

We acknowledge the sample size is relatively small and that our study was not blinded. However, the current study has some points of strength, given the limited published literature on the topic. The study found that dedicated counseling services provided by community pharmacists to patients with parasitic diseases significantly promoted good hygiene and reduced parasitic infection recurrence.

The deworming service is complex and requires sufficient staffing, resources, time dedicated to the service, laboratory access, further training, ongoing teamwork communication, and logistical considerations to ensure optimal, accessible, and convenient care. Nevertheless, this study showed that there is scope for better improvements to ensure community pharmacy-led health services are providing optimum care. It is recommended that the impact of a specific deworming continuing professional development training program on community pharmacists be studied.

## 5. Conclusions

This trial showed that pharmacist-delivered counseling, reinforced by diligent personal hygiene, decisively curtailed the recurrence of parasitic infections, whereas age in early school years posed the greatest risk and adulthood offered the lowest. Counselling and hygiene emerged as the only independent predictors of sustained clearance in multivariate analysis; gender, parasite species, and repeat dosing did not materially influence outcomes. These findings highlight the pivotal role of community pharmacists in coupling medication guidance with behavior-based prevention strategies to break the reinfection cycle in resource-constrained settings.

## Figures and Tables

**Figure 1 tropicalmed-10-00215-f001:**
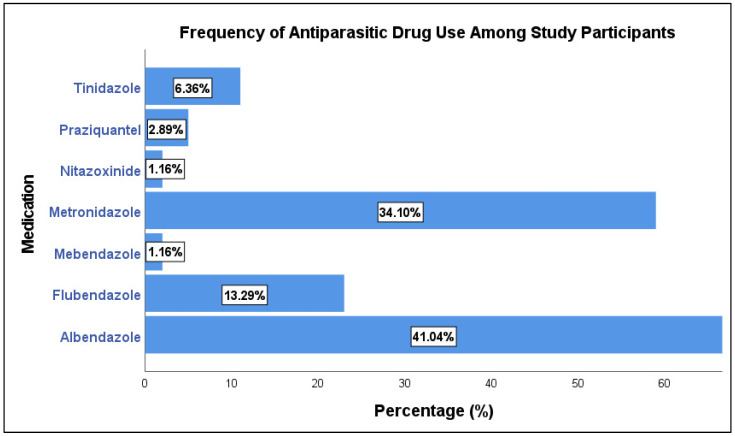
Types of drugs used.

**Figure 2 tropicalmed-10-00215-f002:**
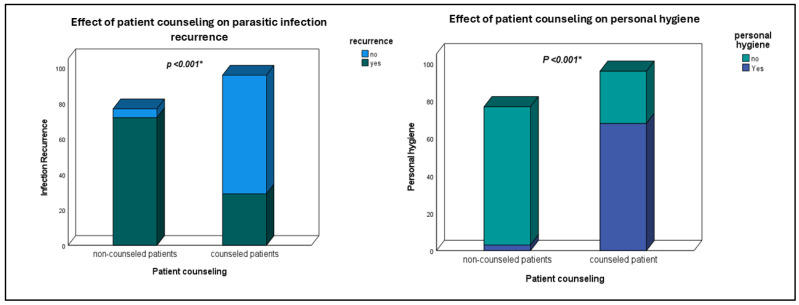
Stacked bar chart showing the effect of patient counseling on infection recurrence and personal hygiene. *: statistically significant (*p* < 0.05).

**Table 1 tropicalmed-10-00215-t001:** Demographic and clinical characteristics of participants (N = 173).

Parameter	N (%)
Age, years	
0–5	47 (27.2%)
6–11	46 (26.6%)
12–18	29 (16.8%)
>18	51 (29.5%)
Gender	
Female	92 (53.2%)
Male	81 (46.8%)
Type of infection	
*Oxyuris*	77 (44.5%)
*Entamoeba*	50 (28.9%)
*Entamoeba Cyst*	15 (8.7%)
*Ascaris*	15 (8.7%)
*Schistosomiasis*	5 (2.9%)
*Giardia*	8 (4.6%)
*Pinworm*	3 (1.7%)
Personal hygiene	
Yes	71 (41%)
No	102 (59%)
Patient counseling	
Yes	96 (55.5%)
No	77 (44.5%)
Dose repeat	
Yes	119 (68.8%)
No	54 (31.2%)
Recurrence	
Yes	101 (58.4%)
No	72 (41.6%)

Data is represented as a number (percentage).

**Table 2 tropicalmed-10-00215-t002:** Factors affecting recurrence (N = 173).

Parameter	Recurrence		*p* Value
	Yes	No	
Age, years			0.002 *
0–5	26 (55.3%)	21 (44.7%)	
6–11	35 (76.1%)	11 (23.9%)	
12–18	20 (69.0%)	9 (31.0%)	
>18	20 (39.2%)	31 (60.8%)	
Gender			0.402
Female	51 (55.4%)	41 (44.6%)	
Male	50 (61.7%)	31 (38.8%)	
Type of infection			0.018 *
*Oxyuris*	41 (53.2%)	36 (46.8%)	
*Entamoeba*	34 (68.0%)	16 (32.0%)	
*Entamoeba Cyst*	7 (46.7%)	8 (53.3%)	
*Ascaris*	7 (46.7%)	8 (53.3%)	
*Schistosomiasis*	1 (20.0%)	4 (80.0%)	
*Giardia*	8 (100%)	0 (0%)	
*Pinworm*	3 (100%)	0 (0%)	
Personal hygiene			<0.001 *
Yes	16 (22.5%)	55 (77.5%)	
No	85 (83.3%)	17 (16.7%)	
Patient counseling			<0.001 *
Yes	29 (30.2%)	67 (69.8%)	
No	72 (93.5%)	5 (6.5%)	
Dose repeat			0.132
Yes	74 (62.2%)	45 (37.8%)	
No	27 (50.0%)	27 (50.0%)	

Values are presented as *n* (column %). For each variable, the distribution of categories between patients with recurrence (“Yes”) and without recurrence (“No”) was compared using Pearson’s Chi-square test (two-tailed); *p*-values < 0.05 are considered statistically significant and are flagged with *. Reference category for each comparison: Age > 18 years, male sex, *Oxyuris* infection, “Yes” to personal hygiene, “Yes” to patient counseling, and “Yes” to repeat dose. These categories were used as the baseline when calculating Chi-square residuals. This was an exploratory analysis; therefore, no adjustment for multiple testing was applied.

**Table 3 tropicalmed-10-00215-t003:** Relationship between patient counseling and personal hygiene.

	Personal Hygiene		
Patient Counseling	Yes	No	*p* Value
Yes	68 (70.8%)	28 (29.2%)	<0.001 *
No	3 (3.9%)	74 (96.1%)	

Data is represented as a number (percentage). * Statistically significant association using Chi-square test *p* < 0.05.

**Table 4 tropicalmed-10-00215-t004:** Evaluation of factors affecting infection recurrence rate using logistic regression.

Variable	Coefficient	Odds Ratio	*p* Value
**Age**			
6–11 (Ref)			0.276
0–5	−0.301	0.740	0.625
12–18	−0.117	0.890	0.881
>18	−1.240	0.289	0.086
**Gender**	0.101	1.106	0.828
**Type of Infection**			
***Schistosomiasis (Ref)***			0.979
***Oxyuris***	0.690	1.994	0.621
***Entamoeba***	1.165	3.207	0.402
***Entamoeba cyst***	1.092	2.980	0.451
***Ascaris***	0.846	2.331	0.573
***Giardia***	18.79	14,586	0.999
***Pinworm***	18.68	12,972	0.999
**Personal Hygiene**	1.516	4.554	0.004 *
**Patient Counselling**	2.536	12.624	<0.001 *
**Dose Repeat**	−0.724	0.485	0.238
**Constant**	−1.521	0.218	0.313

* Statistically significant association between factor and infection recurrence using logistic regression *p* < 0.05.

## Data Availability

Availability of Data and Materials: Data are available from the corresponding author (H.A.M.M) upon reasonable request.

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
