# Peer review of "The Impact of a Community Pharmacy-Led Deworming-Related Counselling Service: An Interventional Study in a Low-to-Middle Income Country"

_tropicalmed, 2025, doi:10.3390/tropicalmed10080215_

Round 1
Reviewer 1 Report
Comments and Suggestions for Authors
The authors are addressing a real public health gap. I appreciate the effort and community-based focus. That said, there are a few areas where the manuscript would benefit from greater clarity, particularly around methodology and variable definitions. The comments below will strengthen the manuscript:
- It would be helpful to know if the pharmacists providing the counselling were given any standardized training beforehand. Also, were the same pharmacists involved with both the control and intervention groups? That could introduce bias or contamination, so it’s worth clarifying.
- How was treatment handled in the control group? In Section 2.4, you describe the counselling approach for the intervention group, but it’s not clear how drugs were prescribed or explained to the control group. Did they get standard pharmacist advice or just a prescription from a physician? Some context here would help readers understand what "usual care" looked like.
- How was recurrence measured? Was there a follow-up stool test, or was this based on self-reported symptoms? If it’s self-report, that’s okay, but it should be made clear, and the potential limitations acknowledged.
- The manuscript presents “personal hygiene” and “patient counselling” as two distinct variables (e.g., Table 1 and Line 245), yet the intervention description (Lines 193–200) clearly states that hygiene education was part of the pharmacist-led counselling. This creates confusion: if hygiene promotion was included in the counselling sessions, how can “personal hygiene” be independently reported without specifying its relationship to counselling? In Line 245, the authors state: “Personal hygiene practices were reported by 71 participants (41%)” — but it is unclear how these practices were defined or assessed. Was this self-reported? Based on a checklist? Observed behaviours? Additionally, was this measurement taken before or after the counselling intervention? This ambiguity persists in Line 260, where the authors state: “Personal hygiene practices were significantly increased” — without detailing how this increase was measured, when it occurred, or relative to what baseline. To strengthen the manuscript’s internal consistency and analytical validity, the authors should: Clearly define what constitutes “personal hygiene practices” and how these were operationalized and evaluated. Specify the timing of this measurement (e.g., pre- vs. post-intervention). Clarify whether participants in the control group were also assessed for personal hygiene and how they received hygiene information, if at all. Reconcile the distinction between “hygiene” and “counselling” throughout the Results and Tables to avoid apparent overlap or circular reasoning in statistical associations.
- Please revise Table 2legend to clearly: Define the comparison groups for each variable; Indicate the reference category where applicable; Specify the exact test used; If multiple comparisons were made, indicate whether any correction was applied.
- Line 61 – Please define NTD. Spell out “Neglected Tropical Diseases” the first time you use the abbreviation.
- Please italicize all microorganism names (e.g., Entamoeba histolytica), and use the full genus name the first time you mention each species.
- Line 96 – Be more specific about genes. The line “Factors related to the host, such as gene expression, play a role in invasive amebiasis” feels a little vague. Could you briefly mention the type of genes involved (e.g., immune response, epithelial barrier)?
Author Response
Comment 1: The authors are addressing a real public health gap. I appreciate the effort and community-based focus. That said, there are a few areas where the manuscript would benefit from greater clarity, particularly around methodology and variable definitions. The comments below will strengthen the manuscript:
Response 1: Thank you for your great effort exerted in helping us improve the clarity and the value of our manuscript.
Comment 2: It would be helpful to know if the pharmacists providing the counselling were given any standardized training beforehand.
Response 2:We thank the reviewer for this insightful comment. A standardized training program was conducted for the pharmacists involved in the intervention group prior to the study. This training focused on consistent delivery of counselling content, including drug use, hygiene practices, and adherence strategies, based on a prepared counselling protocol. This has now been clarified and highlighted in the revised methods section 2.2 .
Comment 3:Also, were the same pharmacists involved with both the control and intervention groups? That could introduce bias or contamination, so it’s worth clarifying.
Response 3:To avoid bias, different pharmacists were assigned to the intervention and control groups. Those providing counselling services were not involved in dispensing medications to the control group participants. We have now added a statement to the manuscript to in methods section 2.2 explicitly describe this separation.
Comment 4: How was treatment handled in the control group?
Response 4:Thank you for pointing this out. Participants in the control group received standard care, which involved physician-prescribed medications without additional structured counselling. Any communication from pharmacists was limited to basic dispensing instructions, with no additional educational or adherence guidance provided. This clarification has now been added to Section 2.4.
Comment 5:In Section 2.4, you describe the counselling approach for the intervention group, but it’s not clear how drugs were prescribed or explained to the control group. Did they get standard pharmacist advice or just a prescription from a physician? Some context here would help readers understand what "usual care" looked like.
Response 5: We acknowledge this important notice. Participants in the control group received standard care; physician-prescribed medications without additional structured counselling. Any communication from pharmacists was limited to basic dispensing instructions, with no additional educational or adherence guidance provided. This clarification has now been added to Section 2.4.
Comment 6:How was recurrence measured? Was there a follow-up stool test, or was this based on self-reported symptoms? If it’s self-report, that’s okay, but it should be made clear, and the potential limitations acknowledged.
Response 6: Thank you so much for your comment. Recurrence was measured based on self-reported symptoms and rebeated stool analysis within a follow-up period of four weeks. This is mentioned in methods section.
Comment 7:The manuscript presents “personal hygiene” and “patient counselling” as two distinct variables (e.g., Table 1 and Line 245), yet the intervention description (Lines 193–200) clearly states that hygiene education was part of the pharmacist-led counselling. This creates confusion: if hygiene promotion was included in the counselling sessions, how can “personal hygiene” be independently reported without specifying its relationship to counselling?
Response 7:We have revised both the intervention description and Results sections to clarify this relationship and now explicitly state that improved hygiene practices were an expected outcome of the counselling. This also helps distinguish between exposure (intervention) and behavioral outcomes.
Comment 8:In Line 245, the authors state: “Personal hygiene practices were reported by 71 participants (41%)” — but it is unclear how these practices were defined or assessed. Was this self-reported? Based on a checklist? Observed behaviours? Additionally, was this measurement taken before or after the counselling intervention?
Response 8:In response, we have revised the Methods section (now Section 2.4) to provide a clear definition of “personal hygiene practices.” These were assessed using a structured, self-administered checklist comprising eight items, including:
- Washing hands with soap and water after using the toilet
- Washing hands with soap and water before and after eating.
- trimming fingernails regularly.
- Drinking clean, safe water (filtered/boiled/bottled)
- Change and wash bed sheets regularly.
- Washing fruits and vegetables thoroughly before eating.
- Defecating in a sanitary toilet, not performing open defecation
- Educating my children/family members about handwashing and hygiene
Participants were asked to indicate whether they practiced each behavior consistently (“Yes”/“No”) at two time points at baseline (pre-intervention) and at follow-up (12 weeks post-intervention). The score was then converted into a binary outcome (adequate vs. inadequate hygiene) for analysis. This definition is now clearly stated in both the Methods and Results sections.
Comment 9:This ambiguity persists in Line 260, where the authors state: “Personal hygiene practices were significantly increased” — without detailing how this increase was measured, when it occurred, or relative to what baseline.
Response 9:We acknowledge this ambiguity and have now clarified that hygiene practices were measured at two time points at baseline (pre-intervention) and at follow-up (12 weeks post-intervention). The measurement assessed the difference between pre and post intervention between the two group. This timing has been explicitly stated in the methods section 2.4
Comment 10:To strengthen the manuscript’s internal consistency and analytical validity, the authors should: Clearly define what constitutes “personal hygiene practices” and how these were operationalized and evaluated. Specify the timing of this measurement (e.g., pre- vs. post-intervention).
Response 10:Thank you for this comment. Now we clarified that hygiene practices were measured at two time points at baseline (pre-intervention) and at follow-up (12 weeks post-intervention). This timing has been explicitly stated in the methods section 2.4 .
Comment 11:Clarify whether participants in the control group were also assessed for personal hygiene and how they received hygiene information, if at all. Reconcile the distinction between “hygiene” and “counselling” throughout the Results and Tables to avoid apparent overlap or circular reasoning in statistical associations.
Response 11: We have clarified that personal hygiene practices were assessed in both the control and intervention groups at baseline and follow-up. However, participants in the control group did not receive structured hygiene education, and any change in their hygiene behavior was based solely on general public awareness or prior knowledge. This comparison allows us to attribute observed differences more confidently to the intervention.
Comment 12:Please revise Table 2legend to clearly: Define the comparison groups for each variable; Indicate the reference category where applicable; Specify the exact test used; If multiple comparisons were made, indicate whether any correction was applied.
Response 12: Thanks, we appreciate your comment and suggestion. We revised the table 2 legend as requested.
“Values are presented as n (column %). For each variable, the distribution of categories between patients with recurrence (“Yes”) and without recurrence (“No”) was compared using Pearson’s Chi‑square test (two‑tailed); p‑values < 0.05 are considered statistically significant and are flagged with *. Reference category for each comparison: Age > 18 years, Male sex, Oxyuris infection, “Yes” to personal hygiene, “Yes” to patient counselling, and “Yes” to repeat dose. These categories were used as the baseline when calculating Chi‑square residuals. This was an exploratory analysis; therefore, no adjustment for multiple testing was applied.”
Comment 13: Line 61 – Please define NTD. Spell out “Neglected Tropical Diseases” the first time you use the abbreviation.
Please italicize all microorganism names (e.g., Entamoeba histolytica), and use the full genus name the first time you mention each species.
Response 13:We appreciate the reviewer’s valuable suggestion. We spelled out Neglected Tropical Diseases the first time it appeared, italicized the microorganism names, and used the full genus name the first time we mention each species, as per your recommendation.
Comment 14: Line 96 – Be more specific about genes. The line “Factors related to the host, such as gene expression, play a role in invasive amebiasis” feels a little vague. Could you briefly mention the type of genes involved (e.g., immune response, epithelial barrier)?
Response 14:We added the required details as per your recommendation.
Reviewer 2 Report
Comments and Suggestions for Authors
This study examines the association of community pharmacist-provided patient counseling with the recurrence of parasites and personal hygiene practices among Egyptians residing in a rural community. The study addresses an important issue affecting morbidity and mortality among individuals experiencing poverty, particularly those residing in middle- and low-income nations. The paper is interesting in its content. The study has an appropriate design, and the results and discussion are presented well. The tables and figures are appropriate.
I have a few suggestions.
- Including sub-section headings will improve the readability of the Introduction needs sub-
- Section 2.4 describes procedures for participants in the Intervention group. (a) The authors need to indicate the usual care that participants in the control group received. (b) The authors mention in Section 2.1 that the intervention included two follow-up phone calls; they need to describe the content of these phone calls in Section 2.4. They need to specify the percent of the intervention participants who received no follow-up calls, one follow-up call, or two follow-up calls.
- The authors need to define the measures used in the analysis.
- Line 245: “Personal hygiene practices were reported by 71 participants (41%).” What does this mean? Were only 41% of participants asked about personal hygiene practices? Did 41% actually use personal hygiene practices; if so, what were these personal hygiene practices? As noted above, the authors need to define the measures used in the analysis.
- Further, what do “dose repeat” and “recurrence” mean? Did some participants get a second dose during the 12 month intervention? How did intervention and control groups compare in terms of dose repeat? What drugs were repeated?
Author Response
Comment 1:This study examines the association of community pharmacist-provided patient counseling with the recurrence of parasites and personal hygiene practices among Egyptians residing in a rural community. The study addresses an important issue affecting morbidity and mortality among individuals experiencing poverty, particularly those residing in middle- and low-income nations. The paper is interesting in its content. The study has an appropriate design, and the results and discussion are presented well. The tables and figures are appropriate.
Response 1: We appreciate the reviewer’s valuable suggestions and the time and effort exerted to enhance our manuscript
Comment 2:I have a few suggestions. Including sub-section headings will improve the readability of the Introduction
Response 2:Thanks. We added the recommended sub-section headings in the introduction.
Comment 3:Section 2.4 describes procedures for participants in the Intervention group. (a) The authors need to indicate the usual care that participants in the control group received.
Response 3:Thank you for your recommendation. We appreciate this important advice. Participants in the control group received standard care, which involved physician-prescribed medications without additional structured counselling. Any communication from pharmacists was limited to basic dispensing instructions, with no additional educational or adherence guidance provided. This clarification has now been added to Section 2.4.
Comment 4:(b) The authors mention in Section 2.1 that the intervention included two follow-up phone calls; they need to describe the content of these phone calls in Section 2.4.
Response 4:Thanks. We have now expanded Section 2.4 to describe each call’s timing and structured content. Specifically, Call 1 (week 2) confirms treatment initiation, screens for adverse effects, resolves practical barriers, and re‑emphasises the eight hygiene behaviours, while Call 2 (week 8) verifies course completion, reassesses hygiene adherence, screens for recurrent symptoms, and reminds patients of their 12‑week stool‑analysis visit. These additions align our description with TIDieR guidelines and improve the reproducibility of the intervention.
Comment 5:They need to specify the percent of the intervention participants who received no follow-up calls, one follow-up call, or two follow-up calls.
Response 5:We appreciate your suggestion to quantify call completion. Section 2.4 now reports that 90 of 96 intervention participants (93.8 %) completed both follow‑up calls, 4 participants (4.2 %) completed only one call, and 2 participants (2.1 %) could not be reached despite three attempts and therefore received no calls. Including these figures clarifies intervention fidelity and facilitates interpretation of our outcomes.
Comment 6:The authors need to define the measures used in the analysis.
Response 6:Thank you for requesting clearer outcome definitions. We have added explicit operational definitions in Section 2.4: (i) recurrence is defined as symptom re‑appearance plus laboratory confirmation of the same parasite within 12 weeks; (ii) adequate hygiene denotes affirmative responses to at least six of the eight checklist items at week 12; and (iii) dose repeat refers to any additional anthelmintic dose beyond the initial course, verified by pharmacy records or self‑report. Section 2.6 has been updated to note that these binary variables underpin both Pearson’s χ² tests and the multivariable logistic‑regression model. These clarifications align with CONSORT recommendations and enhance the transparency of our analytic approach.
Comment 7:Line 245: “Personal hygiene practices were reported by 71 participants (41%).” What does this mean?
Response 7:In response, we have revised the Methods section (now Section 2.4) to provide a clear definition of “personal hygiene practices.” These were assessed using a structured, self-administered checklist comprising eight items
Comment 8:Were only 41% of participants asked about personal hygiene practices? Did 41% actually use personal hygiene practices; if so, what were these personal hygiene practices? As noted above, the authors need to define the measures used in the analysis.
Response 8:All patients were asked about personal hygiene practices , but 41 % actually used personal hygiene practices
Those practices were:
In response, we have revised the Methods section (now Section 2.4) to provide a clear definition of “personal hygiene practices.” These were assessed using a structured, self-administered checklist comprising eight items, including:
- Washing hands with soap and water after using the toilet
- Washing hands with soap and water before and after eating.
- trimming fingernails regularly.
- Drinking clean, safe water (filtered/boiled/bottled)
- Change and wash bed sheets regularly.
- Washing fruits and vegetables thoroughly before eating.
- Defecating in a sanitary toilet, not performing open defecation
- Educating my children/family members about handwashing and hygiene
Participants were asked to indicate whether they practiced each behavior consistently (“Yes”/“No”) during the study period. The score was then converted into a binary outcome (adequate vs. inadequate hygiene) for analysis. This definition is now clearly stated in both the Methods and Results sections.
Comment 9:Further, what do “dose repeat” and “recurrence” mean? Did some participants get a second dose during the 12 month intervention? How did intervention and control groups compare in terms of dose repeat? What drugs were repeated?
Response 9:Thank you so much for this comment. Recurrence was measured based on self-reported symptoms and rebeated stool analysis within a follow-up period of 12 weeks. This has now been explicitly mentioned in the pateints and methods (Section 2.3).
Reviewer 3 Report
Comments and Suggestions for Authors
The submitted manuscript should be revised critically as it is written not so impressive way, there are large number of mistakes and issues;
- The introduction is too much large, it should be concise and to the point.
- The graphical abstract should e revised and should be presented with the theme of the manuscript.
- The equations in the FIG 1 should be redrawn by the authors
- Figure 2 should be written accurately and set appropriate
- Fig 3 and Fig 4 should be combine for the better presentation.
- The data in TABLE 4 should be extracted in the main text and remove the this table
- The conclusion should be re-written and focus on main finding of the studies.
Author Response
Comment 1:The submitted manuscript should be revised critically as it is written not so impressive way, there are large number of mistakes and issues;The introduction is too much large, it should be concise and to the point.
Response 1:Thank you for your valuable recommendation. We revised the manuscript, especially the introduction section as per your advice.
Comment 2:The graphical abstract should e revised and should be presented with the theme of the manuscript.
Response 2:Thank you for your valuable feedback regarding the graphical abstract. We have revised it to better align with the theme and focus of the manuscript.
Comment 3:The equations in the FIG 1 should be redrawn by the authors
Response 3: Thank you
Comment 4:Figure 2 should be written accurately and set appropriate
Response 4:Thank you for your observation. I revised Figure 2 to ensure it is accurately labeled, clearly presented, and appropriately reflects the content of the manuscript.
Comment 5:Fig 3 and Fig 4 should be combine for the better presentation.
Response 5:Thank you for your helpful comment. Figures 3 and 4 have been combined into a single figure to improve clarity and presentation.
Comment 6:The data in TABLE 4 should be extracted in the main text and remove the this table
Response 6:Thank you for your suggestion. As requested, we have removed Table 4 and integrated its contents into the main text. The results of the logistic regression analysis, including both univariate and multivariate findings related to age, gender, type of infection, personal hygiene, patient counselling, and dose repetition, have been clearly described in the Results section
Comment 7:The conclusion should be re-written and focus on main finding of the studies.
Response 7:Thanks. We revised it and now it is based on key findings.
Round 2
Reviewer 1 Report
Comments and Suggestions for Authors
The manuscript is ready for publication
Reviewer 3 Report
Comments and Suggestions for Authors
May be accepted